# Implementation and Delivery of Oral Cholera Vaccination Campaigns in Humanitarian Crisis Settings among Rohingya Myanmar nationals in Cox’s Bazar, Bangladesh

**DOI:** 10.3390/vaccines11040843

**Published:** 2023-04-14

**Authors:** Ashraful Islam Khan, Md Taufiqul Islam, Zahid Hasan Khan, Nabid Anjum Tanvir, Mohammad Ashraful Amin, Ishtiakul Islam Khan, Abu Toha M. R. H. Bhuiyan, A. S. M. Mainul Hasan, Muhammad Shariful Islam, Tajul Islam Abdul Bari, Aninda Rahman, Md. Nazmul Islam, Firdausi Qadri

**Affiliations:** 1Infectious Diseases Division, International Centre for Diarrhoeal Disease Research Bangladesh (icddr,b), Dhaka 1212, Bangladesh; ashrafk@icddrb.org (A.I.K.); taufiqulislam@icddrb.org (M.T.I.); nabid.anjum@icddrb.org (N.A.T.); ashraful.amin@icddrb.org (M.A.A.); ishtiakul.khan@icddrb.org (I.I.K.); tajul.islam@icddrb.org (T.I.A.B.); 2School of Medical Science, Griffith University, Gold Coast 4222, Australia; 3Refugee Health Unit, Office of the Refugee Relief and Repatriation Commissioner, Cox’s Bazar 4700, Bangladesh; dr.tohabhuiyan@gmail.com; 4Health Section, United Nations Children’s Fund (UNICEF), Cox’s Bazar 4700, Bangladesh; ashasan@unicef.org; 5Integrated Management of Childhood Illness, Directorate General of Health Services, Dhaka 1212, Bangladesh; muhammadshariful@gmail.com; 6Communicable Disease Control, Directorate General of Health Services (DGHS), Dhaka 1212, Bangladesh; dr.turjossmc@gmail.com (A.R.); nimunna@yahoo.com (M.N.I.)

**Keywords:** Bangladesh, cholera, Cox’s Bazar, Rohingya Myanmar nationals (RMN), oral cholera vaccination

## Abstract

Background: Over 700,000 Myanmar nationals known as the ‘Rohingyas’ fled into Cox’s Bazar, Bangladesh, in late 2017. Due to this huge displacement into unhygienic areas, these people became vulnerable to communicable diseases including cholera. Assessing the risk, the Government of Bangladesh (GoB), with the help of the International Centre for Diarrhoeal Disease Research, Bangladesh (ICDDR,B) and other international partners, decided to take preventive measures, one of which is the execution of oral cholera vaccination (OCV) campaigns. This paper describes the implementation and delivery of OCV campaigns during humanitarian crises in Bangladesh. Methods: Seven rounds of OCV campaigns were conducted between October 2017 and December 2021. The OCV campaigns were conducted by applying different strategies. Results: Approximately 900,000 Rohingya Myanmar nationals (RMNs) and the host population (amounting to 528,297) received OCV across seven campaigns. In total, 4,661,187 doses of OCVs were administered, which included 765,499 doses for RMNs, and 895,688 doses for the host community. The vaccine was well accepted, and as a result, a high level of coverage was achieved, ranging from 87% to 108% in different campaigns. Conclusions: After successful pre-emptive campaigns in Cox’s Bazar humanitarian camps, no cholera outbreaks were detected either in the RMN or host communities.

## 1. Introduction

The Rohingya people have been forced to migrate to Bangladesh since 1978. These people have faced decades of systematic discrimination, statelessness, and targeted violence in the Rakhine state of Myanmar. Such persecution has forced Rohingya people of all ages and genders into Bangladesh for many years, with significant spikes following violent attacks in 1978, 1991–1992, and again in 2016–2017. According to UN Fact-Finding, a considerable population crossed the border to save their lives following the occurrence of mass genocide in August 2017 [1]. Over 700,000 Rohingya people fled into Cox’s Bazar, a coastal area in the southern part of Bangladesh, in 2017. The Rohingya refugee crisis has been considered the most extensive and fastest movement of people in recent history [2]. These displaced people lived in densely populated settings, resulting in susceptibility and vulnerability to numerous communicable diseases including cholera. The most common causes of illness were acute respiratory infection (ARI), acute watery diarrhea (AWD), and undifferentiated fever, all resulting from the poor sanitary and unhygienic living conditions of these people.

Cholera is an acute watery diarrheal disease caused by toxigenic strains of the bacterium *Vibrio cholerae*. It causes over 3 million cases and approximately 100,000 deaths annually in countries in which cholera is endemic, and frequent epidemics in other areas with a lack of water and sanitation infrastructure [3]. The disease is characterized by the acute onset of watery diarrhea, which leads to rapid dehydration and death if not promptly treated. It is difficult to use intravenous infusion without adequate medical facilities, particularly in humanitarian settings. Proper water and sanitation infrastructure may play a significant role in epidemic control. It is now apparent that prophylaxis such as vaccines can effectively prevent cholera.

The oral cholera vaccine (OCV) stockpile was established by the WHO for use in emergencies (outbreak control and humanitarian crises) [4] and non-emergencies, and has been in operation since 2013 [5]. The International Coordinating Group (ICG) of the WHO manages the emergency OCV supply, which has been largely allocated for the control of outbreaks and for use in epidemics during humanitarian crises. It has been used in cholera outbreaks in Haiti, Africa, Asia, and the Middle East. Since 2013, more than 36 million doses have been shipped to 24 countries from global stockpiles for 104 OCV campaigns [4]. Bangladesh is an endemic country for cholera, which is a public health concern. Assessing the risk, the Government of Bangladesh (GoB), with the help of the International Centre for Diarrhoeal Disease Research, Bangladesh (ICDDR,B) and other international partners, decided to take preventive measures and conducted seven rounds of OCV campaigns from 2017 to 2021. This paper aims to describes the implementation and delivery of these oral cholera vaccination campaigns during humanitarian crises among the high-risk population of Rohingya Myanmar nationals (RMNs) in Bangladesh.

## 2. Vaccination Area and Population

Vaccinations were conducted at Ukhiya and Teknaf Upazila in Cox’s Bazar District, situated in the Chittagong Division of Bangladesh. The Rohingya people lived in makeshift camps situated in Kutupalong and Balukhali in Ukhiya, and Leda, Nayapara, and Shamlapur in Teknaf Upazila, upon arrival in Bangladesh. According to the population statistics of 2022 conducted by the Government of Bangladesh and the United Nations High Commissioner for Refugees (UNHCR), 920,994 Rohingya people from 193,457 families live in these camps (Figure 1). Among them, 884,595 individuals from 187,134 families were part of new influxes in 2017, while others arrived in the 1990s. Currently, there are 34 refugee camps in Cox’s Bazar, including Bhasan Char, among which 26 are situated in Ukhiya and 7 in Teknaf. Around the camps, the Bangladeshi host population lives in close contact with RMNs. According to data from the Bangladesh 2011 census, the total population of Ukhiya Upazila was 207,379, and the total population of Teknaf Upazila was 264,389. The risks of cholera and other diarrheal diseases are similar for both the Bangladeshi host population and Rohingya Myanmar nationals.

## 3. Strategies for Oral Cholera Vaccination Campaigns

Cholera control and prevention are considered priorities in emergency settings, especially in endemic zones. Some mainstays of control measures to be implemented during emergencies should be (i) the improvement of water and sanitation, (ii) appropriate treatment for people with cholera, and (iii) mobilizing communities. All of these measures were supported by epidemiological surveillance. However, vaccination should not disrupt the provision of other high-priority health interventions to control or prevent cholera. Vaccines provide immediate short-term protection. Interventions to improve access to safe water and sanitation should be implemented. The Bangladesh government has delivered over four million doses of OCV (Shanchol^TM^ and Euvichol- Plus) to the Rohingya Myanmar nationals and the host communities in Ukhiya and Teknaf since the massive influx of Myanmar nationals from August 2017 [6]. There were seven rounds of campaigns for delivering the OCV, and different strategies were followed during the campaigns, which consisted of two OCV dose schedules.

## 4. First and Second-Round Campaigns: Fixed Site Strategy–Mass Campaign

The GoB submitted a request to the International Coordinating Group (ICG) for the emergency use of OCV in challenging conditions among Rohingya Myanmar nationals on 27 September 2017. The ICG responded to the request within 24 h and approved 900,025 doses of OCV (Shanchol^TM^) for the campaign with the support of Gavi. The Government of Bangladesh prepared a strategic plan, and the first round of vaccination was between the 10th and 18th November, 2017, targeting 658,371 RMNs aged ≥1 year. As OCV is not contraindicated during pregnancy [6], pregnant women were not excluded from this vaccination campaign. In this OCV delivery, 205 vaccination teams were deployed during the campaign period at the camp level [7]. Among them, 150 teams were deployed in Ukhiya, and the remaining 55 were in Teknaf Upazila. Each team had six members: two vaccinators (who administered the vaccine), two mobilizers (who mobilized people from camps and maintained queues at the vaccination site), one record-keeper, and one person who marked the vaccine recipient’s finger with gentian violet ink. Local Rohingya volunteers and a ‘Majhi’ who acted as a leader (one member of the six-member team) in camps were included in the micro-plan to make this campaign successful and acceptable to the community. The Majhi’s role was to motivate and bring people to vaccination sites. They also actively participated in creating awareness in the community. Vaccines were delivered from a common location at a central point to cover a few vaccination sites, and each team returned the unused vaccines at the end of the day. In this campaign, fixed sites for vaccine delivery to beneficiaries were established in a suitable place, such as the house of the Majhi, learning centers, and distribution points that were accessible to everyone; these sites were identified easily by the beneficiaries, as the Moni flag was placed in every vaccination site during the campaign days. One soap for each beneficiary was distributed among the vaccine recipients. The second round of OCV campaigns was conducted from the 4 to the 9 of November, 2017, targeting approximately 180,000 children aged 1–5 years, for added protection. This campaign followed the same strategy as the first round of the OCV campaign. In the first and second rounds of the OCV campaign, a vaccine card was not delivered to the recipient. Campaigns were conducted through a joint effort by the government of Bangladesh, international agencies [United Nations International Children’s Emergency Fund (UNICEF), WHO, Médecins Sans Frontières, International Organization for Migration, and the ICDDR,B and non-governmental organizations. Along with OCV, the government of Bangladesh also administered OPV and MR vaccines during the second round of the OCV campaign to control outbreaks of measles, rubella, and poliomyelitis [8]. During the campaign, health education materials, such as banners and leaflets focusing on cholera control behaviors, were distributed.

## 5. Third-Round Campaign: Fixed Site and Mobile Team Approach

In February 2018, the ICG approved 984,906 doses of OCV from the global stockpile in response to the second application of the GoB. The third round of the OCV campaign was conducted in Rohingya camps from 6 to 13 May 2018. In this campaign, a second dose of OCV was administered to Rohingya people aged >5 years who received one dose of OCV from an earlier campaign conducted in October 2017, and a first dose was administered to age-migrating children from 12 months to 18 months and newly arrived Rohingya Myanmar nationals. The host population of the area adjacent to the camps was also included in the round. A total of 259 teams participated in the campaign, and each team consisted of six members, including a Majhi, as mentioned in the first two campaigns [9]. The only difference from the last round was that there were no finger markings. A few team members went to a common gathering place, acting as a mobile team, to cover people who did not come to the fixed vaccination site. A vaccine card was introduced in this round, with two different colors for vaccine recipients: a white-colored vaccine card for Rohingya Myanmar nationals and a yellow card for the host community. A health education message was also circulated during the campaign, which focused on handwashing, food hygiene, and safe water.

## 6. Fourth-Round Campaign: Using a Routine Immunization Platform

The GoB submitted a third request to the ICG in October 2018 and received 328,556 doses of OCV. The fourth round of the OCV campaign was conducted with the help of the remaining OCV stocks in Cox’s Bazar, between 17 November and 23 December 2018. The Kutupalong Community Clinic and Balukhali Health sub-Center were used as the vaccine distribution points. In this campaign, 70 mobile outreach teams (two members in each team) and 46 fixed site teams were used to conduct the campaign and deliver OCV using a routine immunization platform over four weeks [10]. Each team consisted of two vaccinators and two volunteers. Majhis and Rohingya volunteers were enlisted to assist in mobilization during the campaign. Vaccine cards were distributed to all OCV recipients.

## 7. Fifth- and Sixth-Round Campaigns: Based on House-to-House Strategy

During the earlier four campaigns, most of the host population in Ukhiya and Teknaf remained unvaccinated and vulnerable to cholera. In response to the fourth application, the ICG approved 1,270,170 OCV doses. In the first shipment, 635,110 doses were sent to Bangladesh from the global stockpile in December 2019, and in the second shipment, 441,840 doses were sent to Bangladesh.

The fifth round OCV campaign was successfully conducted in the host community between 8 and 31 December 2019. The target population for the host community was all age groups 1 year and above, and for Rohingya Myanmar nationals, the target group was 1 to <5 years of age and the new influx of 1,162 in the camps. The host community target populations in Ukhiya and Teknaf were 239,739 and 255,419, respectively. The OCV campaign in the host community followed a house-to-house strategy with mobile and sweeping teams. In the house-to-house vaccination strategy, 50 mobile teams and 50 sweeping teams worked in each subdistrict. Mobile teams visited every house in their allocated target area according to the micro-plan and vaccinated every person (more than 1 year of age) with one dose of OCV. Each mobile team consisted of five people:one health education messenger, one vaccinator, one vaccine card writer, one tally marker, and one local mobilizer (who belongs to a specific community. The following day, a sweeping team composed of one volunteer and one local mobilizer attempted to administer the vaccine among those who remained unvaccinated on the previous day. Each team visited approximately 60–80 houses per day and targeted 300–350 doses/team/day. The community clinics/FWC/union sub-centers were used to deliver the vaccines as a static site for beneficiaries who missed the vaccine during the campaign and sweeping day.

In the fifth and sixth rounds, two doses of OCV were delivered from 8 to 14 December 2019, and 15 to 20 February 2020. These campaigns covered all the Rohingya Myanmar National children aged less than five years, following the same strategy as the host community campaign.

For all campaigns, on the first four days, each team covered 40–45 households per day to meet the target. They conducted a mop-up activity to vaccinate missed children in the following two days. A sticker was placed in every household during the campaigns, containing information about the total number of household members and the number of vaccinated members. This helped the sweeping team to identify any persons who missed vaccination. This strategy was used in the fifth and sixth round of the OCV campaign for both the Rohingya population and host community. Health education materials (leaflets and infographics) were used to build awareness about cholera control.

## 8. Seventh-Round Campaign: Camp-by-Camp Rolling Approach Strategy

The government of Bangladesh planned to vaccinate all of the Rohingya population with two doses (first dose: 10–24 October 2021; second dose: 1–14 November 2021) of OCV in a seventh round of the campaign, as case numbers rose during this period. This OCV campaign followed camps using a camp-by-camp rolling approach strategy. In this strategy, the team moved to the next camp after the completion of the vaccination of one camp. Two types of OCV vaccines were used in this round; one was Shanchol^TM^ and another one was Euvichol-Plus. Each day, vaccines were deployed according to the micro-plan to the selected camps by maintaining a cold chain (2–8 degree). A specific date for vaccination of each camp was shared with all participants before starting the campaign through loudspeaker announcements (Miking), community health workers (CHWs), communication for development (C4D) volunteers, Majhis and religious leaders, posters, banners, and festoons. Parents and caregivers were encouraged to visit the nearest vaccination site along with their children to receive the vaccine. The target age group was the population above one year of age, including pregnant women and lactating mothers. The total target population was 869,095 (Ukhiya: 712,832 and Teknaf: 156,263). Approximately 139,000 Shanchol^TM^ vaccines were used for both first and second doses in Teknaf.

A total of 250 teams worked in this campaign: 200 for Ukhiya and 50 for Teknaf. Each team had three members: one vaccinator and two volunteers. The vaccinator vaccinated the beneficiaries and performed tally marking. One volunteer was recruited from the host population, as they were permitted to move from one camp to another. They were responsible for providing vaccine cards and finger markings. They used gentian violet or an indelible marker to mark the left little finger of the recipient after each dose. Another volunteer was recruited from the Rohingya Myanmar Nationals and was responsible for controlling the crowd. Vaccinators and volunteers who were recruited from the host community were on the same team, but the volunteers who were recruited from the Rohingya population changed every day, as they were not allowed to move from one camp to another. Each team was supported by a community health worker (CHW) and a Majhi. After each day of vaccination, each team submitted their report to their respective supervisor according to the tally sheet. One supervisor was assigned to each team. Fifty supervisors worked for this campaign.

## 9. OCV Storage/Distribution, Cold Chain

All OCV shipments were stored at the Expanded Program on Immunization (EPI) center in Cox’s Bazar prior to the campaign. Then, OCVs were sent with refrigerated boxes to the Upazila health complex of Ukhiya and Teknaf. Medical technologists (MT) from the EPI of both Upazila health complexes delivered vaccines through carriers (~400 vaccines/carriers), which were transported to the distribution point for delivery wherein one team used one carrier. Throughout the first six rounds, the Shanchol^TM^ vaccine was used to vaccinate both the Rohingya and host populations, which did not require a cold chain on the campaign day. In the seventh round, Euvichol-Plus was delivered, for which a cold chain had to be maintained. Each vaccine carrier contained approximately 300 doses of the Euvichol-Plus vaccine, and cold boxes containing OCV were positioned at the distribution sites.

## 10. AEFI, Monitoring and Evaluation

An adverse event following immunization (AEFI) is any untoward medical occurrence that follows immunization and which does not necessarily have a causal relationship with the usage of the vaccine. The adverse event may be any unfavorable or unintended sign, abnormal laboratory finding, symptom or disease [11]. Vaccine recipients were requested to stay at the vaccination site for follow-up on any immediate adverse events. In addition, all recipients were advised to visit the nearest health facility if any untoward health events occurred after vaccination. Monitoring and evaluation teams were assigned to each campaign. The team’s findings were discussed in coordination meetings to overcome these issues. The monitoring and evaluation teams also collected information on adverse events from a low percentage (1st round: 0.002% and 2nd round: <0.001%) of vaccine recipients.

## 11. Discussion

In the first round of the OCV campaign, 700,487 Rohingya Myanmar nationals were vaccinated, and the vaccine coverage was 106%, as the target population was greater than was estimated. The remaining doses of OCV were maintained for <5 years to complete the two dose regimens. The second round of the OCV campaign was conducted in November, 2017 and the second dose of OCV was delivered to 199,472 children aged 1–5 years [5]. A total of 879,273 patients (89% of the target population) received OCV in the third round. Among them, 775,666 were from the Rohingya population and 103,605 were from the host community [12]. In the fourth round, a total of 356,202 doses were delivered to Rohingya children aged 12–23 months, and a second dose to the Rohingya population and host community who received the first dose in an earlier campaign; the coverage of this campaign was 108% [13].

Host communities (all age groups above 1 year) and the Rohingya population from 1–5 years of age were vaccinated in the fifth and sixth round. In host communities, the OCV campaign covered 251,973 people in Ukhiya and 276,324 people in Teknaf [14]. The campaign coverage was 108%. Due to the lower supply of OCV in the second shipment in January 2020, the OCV campaign was partly conducted in two out of ten unions in Ukhiya and Teknaf Upazila, and 164,615 members of the host population received a second dose of OCV in the sixth round. In refugee camp settings, two rounds of OCV were conducted in December 2019 and February 2020, covering all Rohingya children aged less than five years. The fifth round of the OCV campaign conducted in December 2019 covered a total of 162,871 children (1–5 years) in the Rohingya camps of Ukhiya and Teknaf (11). The sixth-round OCV campaign was completed at the camps in February 2020. During the 6-day campaign, a total of 179,891 beneficiaries were vaccinated [15]. In the seventh round of the OCV campaign, the target population was 869,095. A total of 754,172 Rohingya Myanmar nationals received their first dose, with 87% coverage. Later, 735,907 Rohingya Myanmar nationals received their second dose of OCV, and the coverage of the second dose was 98% (Table 1).

The oral cholera vaccine is considered an important public health tool for controlling both epidemic and endemic cholera globally. Seven large campaigns were carried out in Cox’s Bazar, Bangladesh, both among Rohingya Myanmar nationals and the neighboring host communities. Based on this campaign experience, we can say that OCV delivery is feasible in complex refugee settings. The vaccine was well accepted, and a high level of coverage was achieved. Two reports revealed that coverage of the OCV campaign conducted in Cox’s Bazar among Rohingya Myanmar nationals and the host populations was around/over a hundred percent, which is similar to the OCV campaign conducted in Mozambique in 2016 and Cameroon in 2019 [16,17]. After the successful pre-emptive campaign in the Cox’s Bazar humanitarian camps, no cholera outbreaks were detected [18]. This was possible because of the successful OCV campaigns along with extensive water, sanitation and hygiene (WASH) interventions throughout the time. Diarrhea treatment centers have been established for clinical management of suspected or confirmed cases [19]. The establishment of sentinel surveillance by the ICDDR,B with funding support from UNICEF in different health facilities in collaboration with the government, NGOs, and different international organizations (WHO, IOM, MSF, Red Cross) has helped in the early detection of cholera cases [20]. The joint assessment team (JAT team), which consisted of WASH, health, and laboratory personnel, was formed and acted as a rapid response team for outbreak investigation and active case searching. Each team included epidemiologists and health personnel from the WHO and other organizations, members from the WASH sector, and one member of laboratory personnel from the ICDDR,B. All these interventions helped prevent any potential cholera outbreaks in the makeshift camps. In 2019, an upsurge in cholera cases was observed in both host communities and RMNs, and 239 cases were detected as either RDT-positive or culture-confirmed cases [21]. After the fifth and sixth rounds of the OCV campaign, the number of acute watery diarrhea cases declined in 2020 (15). A similar declining trend in cholera cases was also observed after the OCV campaigns in the Democratic Republic of Congo, Cameroon, and Uganda [22]. There were some limitations to the campaigns. First, the same target population could not be established before the campaign, so the coverage was calculated over a hundred percent. Second, due to vaccine shortages, the entire population, including RMNs and the host community, could not be covered concurrently with two doses of OCV, posing the risk of an increase in cases due to transmission among the unvaccinated.

The International Coordination Group (ICG), which controls emergency supplies of vaccinations, said on 19 October 2022 that a global scarcity of oral cholera vaccines had prompted a change to a one-dose plan, instead of two doses, amid a dramatic spike in outbreaks around the world. According to the World Health Organization (WHO), a two-dose cholera vaccine regimen provides protection against cholera for three years if the second dose is given within six months of the first. The WHO has stated that the single-dose vaccine’s predicted 6-month protective window will enable more individuals to be vaccinated across the globe. The Global Task Force on Cholera Control, a combination of UN agencies, non-governmental organizations, and academic groups, predicted in 2021 that the world will require approximately 250 million doses of cholera vaccine until 2025, both to avoid outbreaks and for immunization programs. Therefore, if any surge is detected in the Rohingya camps, additional reactive campaigns may be needed based on vaccine availability and risk assessment.

## 12. Conclusions

The OCV campaign is feasible in the complex humanitarian crisis settings. No major outbreak was detected in the Rohingya camps, which reflects that cholera epidemics can be prevented in complex conditions by implementing OCV along with other interventions such as WASH.

## 13. Future Direction

Further work is needed to develop a cholera vaccine that is both easy to use and affordable for large-scale use. In addition to vaccination, the WASH intervention, sustainable surveillance systems, and proper case management systems need to be established through a multisectoral approach to prevent cholera outbreaks in humanitarian crises, refugee camps, and surrounding host communities.

## Figures and Tables

**Figure 1 vaccines-11-00843-f001:**
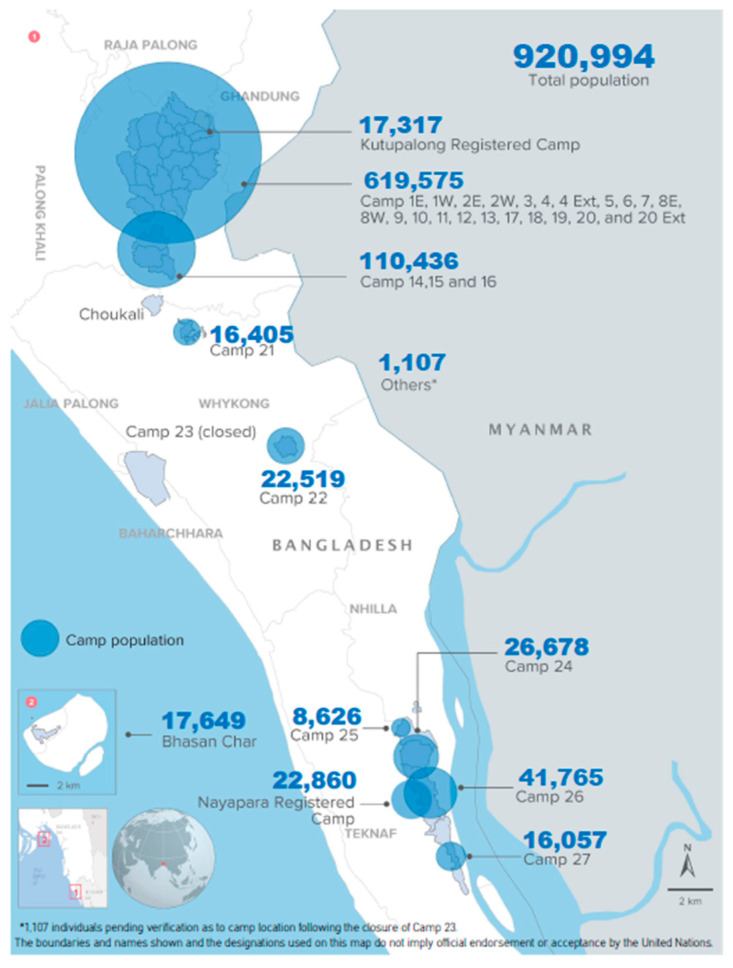
Location camps in Cox’s Bazar (Data source: Joint Government of Bangladesh—UNHCR population Factsheet, 31 January 2022).

**Table 1 vaccines-11-00843-t001:** Vaccination status and target population.

Campaign Round and Dates	Target Population	Doses Delivered
1st Round (10–18 October 2017)	Rohingya Myanmar nationals aged >1 year whole population (1st dose)	700,487
2nd Round (4–9 November 2017)	Rohingya Myanmar nationals aged 1 to ≤5 years (2nd dose)	199,472
3rd Round (6–13 May, 2018)	Rohingya Myanmar nationals > 5 years old, newcomers and above 12 months to 18 months children; Adjacent host community (2nd dose to those in 1st round to those over 5: and also first dose to newcomers in all age and age migrating children; host community (1st dose)	Rohingya Myanmar nationals: 775,666
Host community: 103,605
Total: 879273
4th Round (17 November to 23 December 2018)	Rohingya Myanmar nationals aged 12–23 months and newcomers, 2nd dose	Rohingya Myanmar nationals: 226,955
Host community > 1 year population (2nd dose)	Host community: 99,17
	Total: 356202
5th Round (8–31 December 2019)	Rohingya Myanmar nationals aged from 1 to <5 years vaccinated; Host community >1 year, whole population (1^st^ dose)	Rohingya Myanmar nationals: 162,871
Host community: 528,297
Total: 691,168
6th Round	Host community >1 year, whole population( 2nd dose)	Host community-164,615
(Continued in Host Community) (19–25 January 2020)
Rohingya Myanmar nationals (RMNs) (15–20 February)	Rohingya Myanmar nationals vaccinated: from 1 to <5 years; (2nd dose)	Rohingya Myanmar nationals: 179,891

Total: 344,506
7th Round (10–24 October 2021)	Rohingya Myanmar nationals aged >1 year, whole population (1st dose l)	754,172
7th Round (1–14 November 2021 )	Rohingya Myanmar nationals aged >1 year, whole population (2nd dose )	735,907

## Data Availability

Data can be shared based on the reader’s reasonable request and priority base, and some restrictions will apply.

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
