# Peer review of "Implementation and Delivery of Oral Cholera Vaccination Campaigns in Humanitarian Crisis Settings among Rohingya Myanmar nationals in Cox’s Bazar, Bangladesh"

_vaccines, 2023, doi:10.3390/vaccines11040843_

Round 1

Reviewer 1 Report

This work describes the implementation and delivery of oral cholera vaccination campaigns during humanitarian crises in Bangladesh between October 2017 and December 2021. The major comments are as below:

1. Line 56,99: Please avoid the blank space.

2. Line 69: what does icddr,b stand for?

3. Line 71: Please add to before describe.

4. Line 109: The format of the reference [6] is not consistent with others.

5. Line 164: Please add blank space before respectively.

Author Response

    1. Line 56,99: Please avoid the blank space.

    Response: Thank you for your feedback. We have made the necessary modifications, lines 56, 99 which are also highlighted.

    1. Line 69: what does icddr,b stand for?

    Response: It was previously described in line 69, which is also highlighted.

    1. Line 71: Please add to before describe.

    Response: We have made the necessary modifications, lines 71 which are also highlighted.

    1. Line 109: The format of the reference [6] is not consistent with others.

    Response: Thank you for your feedback. We have made the necessary modifications, line 109 references.

    1. Line 164: Please add blank space before respectively.

    Response: We have made the necessary modifications, line 164 which is also highlighted.

Reviewer 2 Report

Good manuscript describing the implementation of oral cholera vaccine immunization activities. Needs improvement for clarity in several places, but no major concerns. It would be good to see more SIA profiles like this one in the literature.

Needs more context in the introduction. At minimum:
Line 48 - Ought to cite UN findings on Myanmar genocide: https://www.ohchr.org/en/hr-bodies/hrc/myanmar-ffm/index
Line 50 - Citation needed: Considered by whom to be the fastest movement?Line 62 - Citation needed: Examples of cholera vaccine use. Expected durability/utility?

Extra punction or no punctuation in many places. Need to be consistent. 
Line 164 - Using a different numbering system here?

More detail on adverse events would be welcome
Line 220 - Adverse event following immunization not defined anywhere. 
Line 225 - What percentage had data recorded? What was the event rate?

Table 1 is difficult to read; the rows are unclear.

It's worth mentioning/citing the current availability of OCV as part of the discussion. (Vaccine shortage: https://www.thelancet.com/journals/lancet/article/PIIS0140-6736(22)02116-X). Would additional rounds of OCV be possible if presently needed?

Author Response

Line 48 - Ought to cite UN findings on Myanmar genocide: https://www.ohchr.org/en/hr-bodies/hrc/myanmar-ffm/index

Response: Added (lines 47-48)

Line 50 - Citation needed: Considered by whom to be the fastest movement?

Response: We have made the necessary modifications, which are also highlighted.

Line 62 - Citation needed: Examples of cholera vaccine use. Expected durability/utility?

Response: Citation included according to suggestion, which are also highlighted.

Extra punction or no punctuation in many places. Need to be consistent. 

Response: Thank you for your feedback. We have made the necessary modifications, which are also highlighted.

Line 164 - Using a different numbering system here?

Response: We have changed the numbering system, which is also highlighted.

More detail on adverse events would be welcome
Line 220 - Adverse event following immunization not defined anywhere. 

Response: We have made the necessary modifications, Lines: 222-224, which are also highlighted.

Line 225 - What percentage had data recorded? What was the event rate?

Response: The monitoring and evaluation teams also collected information on adverse events from a low percentage (1st round: 0.002% and 2nd round: < 0.001%) of vaccine recipients. We have made the necessary modifications, Lines: 229-230, which are also highlighted.

Table 1 is difficult to read; the rows are unclear.

Response: We have made the necessary modifications

It's worth mentioning/citing the current availability of OCV as part of the discussion. (Vaccine shortage: https://www.thelancet.com/journals/lancet/article/PIIS0140-6736(22)02116-X). Would additional rounds of OCV be possible if presently needed?

Response: Thank you for your feedback. We have made the necessary modifications in the discussion section, lines 282-292 which are also highlighted.
